# Supraglottic airway devices for blind endotracheal intubation: A systematic review

**Jorman H. Tejada-Perdomo**[1,2,3]\*, **Carlos E. Melo-Moreno**[2], **Jesús H. Tovar-Cardozo**[1,2,3], **Jorge A. Ramos-Castaneda**[4,5]

1 Department of Clinical Sciences, Universidad Surcolombiana, Neiva, Colombia, 2 Department of Anesthesiology, Hospital Universitario Hernando Moncaleano Perdomo, Neiva, Colombia, 3 Research Group Desarrollo Social, Salud Pública y Derechos Humanos, Universidad Surcolombiana, Neiva, Colombia, 4 Research Group Innovación y Cuidado, Universidad Antonio Nariño, Neiva, Colombia, 5 Research Group Cuidar, Universidad Surcolombiana, Neiva, Colombia

\* jorman.tejada@usco.edu.co

**Data Availability Statement:** All relevant data are within the paper and its Supporting Information files.

## Abstract

### Introduction

The effectiveness of supraglottic airway devices (SGDs) as a strategy for blind endotracheal intubation (ETI) was compared in this study.

### Methods

A systematic review of clinical trials (CTs) involving SGDs for blind ETI in patients under general anesthesia or simulation manikins, was conducted. CTs that used SGDs for fiberoptic-guided ETI and those conducted in children were excluded. Searches were performed in Embase, MEDLINE (PubMed), Scopus, and LILACS. The primary outcomes examined were the success rate of blind ETI and intubation time. Secondary outcomes were first-attempt intubation success rate and perceived ease of use.

### Results

A total of 567 records were identified from databases, and 16 were identified through citation searches. Ultimately, 27 CTs met the inclusion criteria. The Fastrach Intubating Laryngeal Mask Airway (LMA Fastrach), i-gel, Air-Q Intubating Laryngeal Airway, and Supraglottic Airway Laryngopharyngeal Tube (S.A.L.T.) were the most used SGDs for blind ETI. LMA Fastrach was the most frequently compared device in these CTs. Among the studies in patients, LMA Fastrach and i-gel were the devices that showed the shortest intubation time, although it may be influenced by the way intubation time is assessed. The SGDs with the highest overall success rate were i-gel, S.A.L.T., LMA Fastrach, and single-use LMA Fastrach, followed by Air-Q, and the Intubating Laryngeal Tube Suction-Disposable (iLTS-D2), all achieving success rates greater than 90%. AuraGain had the lowest first-attempt and overall success rates for blind ETI with SGDs.

**Funding:** Publication was funded by Universidad Surcolombiana. The funders had no role in study design, data collection and analysis, decision to publish, or preparation of the manuscript.

**Competing interests:** No authors have competing interests

## Conclusion

New SGDs have not surpassed the LMA Fastrach effectiveness for blind ETI. The single-use LMA Fastrach combines the efficacy of the reusable LMA Fastrach with the features of other SGDs and may be a suitable replacement for them. The I-gel is also a viable alternative for blind ETI, while the AuraGain may not be recommended for this purpose.

## Introduction

Orotracheal intubation is a cornerstone in managing critically ill patients who require a secure airway [1]. However, airway management can sometimes be difficult or beyond the operator's expertise. Due to a high failure rate of approximately 30% with direct laryngoscopy during intubation attempts by paramedics [2], alternative strategies such as supraglottic airway devices (SGDs), videolaryngoscopes, and other blind endotracheal intubation (ETI) devices have been developed to achieve airway isolation. The use of SGDs has been extensively researched and is considered an accepted and recognized technique within the American Society of Anesthesiologists (ASA) difficult airway algorithm [3], particularly when mask ventilation is inadequate after a failed intubation.

The incidence of difficult intubation varies from 0.1% to 10.1% depending on the definition used [4, 5]. In the study by Jayaraj et al., which included 42,805 surgical cases requiring orotracheal intubation, difficult intubation occurred in 0.26% of cases, and failed intubation occurred in 0.03% of cases [6]. The incidence of difficult intubation in patients with difficult intubation criteria ranges from 5.8% to 20% [7–10].

When conventional laryngoscopy fails, SGDs are used as rescue devices to maintain oxygenation and ventilation. Blind ETI using SGDs is an accepted alternative, and several devices facilitate tracheal tube passage [11] with a high success rate [12]. The use of videolaryngoscopy optimized the management of some patients. However, in those whose use was unsuccessful and who required surgical airway management due to the inability to ventilate, problems of poor visualization of the glottis and airway bleeding were observed as causes of failed intubation [6]. Lack of skill in performing rescue surgical procedures or absence of surgical equipment to restore alveolar oxygenation in an emergency will cause anesthesiologists to fail in the primary purpose of securing the airway, which can lead to catastrophic outcomes such as hypoxic brain damage and death [11, 13].

The use of rescue equipment for ventilation and the need for intubation has led to the development of blind intubation devices to meet these airway isolation needs.

Dr. Archie Brain developed the Fastrach Intubating Laryngeal Mask Airway (LMA Fastrach) in 1997 in response to the difficulties of achieving blind intubation with the classic laryngeal mask [14]; since then, it has been used as a strategy for blind ETI with some drawbacks, including the cost of the silicone tube it uses. Various devices, such as laryngeal masks and other direct vision devices, such as fibrobronchoscopes and videolaryngoscopes, have been tested as intubation options to overcome these disadvantages.

Using laryngeal masks is included as an option in difficult airway management. However, there are clinical situations where isolating the airway with orotracheal intubation is important. Studies have reported high success rates for intubation using SGDs [15–19]. In 2018, a meta-analysis reported the success rate of unassisted intubation with SGDs but did not include data on intubation time or studies involving simulation manikins [20]. Therefore, we propose

a systematic review to evaluate the intubation time and success rate of different SGDs as strategies for blind ETI.

## Materials and methods

We developed the protocol for this review in June 2024. This systematic review was conducted according to the recommendations of the Cochrane Collaboration and is presented according to the recommendations of the Preferred Reporting Items for Systematic Reviews and Meta-Analyses (PRISMA) [21].

### Eligibility criteria

Clinical trials (CTs) comparing different SGDs for blind ETI with other intubation devices were included. The studies involved patients under general anesthesia for surgery or simulation manikins. CTs using SGDs for fiberoptic-guided ETI, those conducted in the pediatric population, and those published in languages other than English, Spanish, or Portuguese were excluded. The studies were also excluded if their methods and design were not properly stated or reported or if the data provided was inadequate. Studies were synthesized using a well-formulated Population, Intervention, Control, Outcome (PICO) study design (Table 1).

### Information sources and search strategy

The search was conducted by researchers who have experience in scientific literature review. The authors (JHT-P and CEM-M) independently searched the Embase (1974-June 2024), MEDLINE (PubMed) (1946-June 2024), Scopus (2004-June 2024), and LILACS (1982-June 2024) databases. The search was limited to articles written in English, Spanish, or Portuguese and included studies conducted on humans or simulation manikins. The search strategy was built with the support of an expert in scientific literature search and combined free text and MeSH terms using the Boolean operators AND and OR, as shown below:

((((((("Emergency Medical Services"[Mesh]) OR ("Cadaver"[Mesh])) OR ("Manikins"[Mesh])) OR ("Models, Anatomic"[Mesh])) OR (corpse)) OR ("emergency medical services"[MeSH Terms] OR prehospital emergency care[Text Word])) AND (((blind intubation) OR (SUPRAGLOTTIC DEVICE)) OR (extraglottic devices) OR "Intubation, Intratracheal"[Mesh])) AND ((((("Laryngoscopy"[Mesh] OR "Laryngoscopes"[Mesh]) OR

**Table 1. PICOS statement.**

| Criteria | Determinant |
|---|---|
| Population | Patients under general anesthesia, simulation manikins. |
| Intervention | Blind endotracheal intubation using SGDs |
| Control | Placement of endotracheal tube whit devices other than SGDs. |
| Outcome | Primary outcomes: |
| | Success rate of blind ETI |
| | Blind intubation time whit SGDs |
| | Secondary outcomes: |
| | First-attempt intubation success rate |
| | Perceived ease of use |
| Study design | Clinical trials |

SGDs = supraglottic airway devices.

ETI = endotracheal intubation

(VIDEOLARINGOSCOP*)) OR (video laryngoscopy)) OR (extraglottic devices)) OR (SUPRAGLOTTIC DEVICES))) AND (((((((((((effectiveness) OR ("Efficiency"[Mesh])) OR ("FIRST ATTEMPT")) OR ("FIRST INTENT")) OR ("FIRST TRY")) OR (INTUBATION TIME)) OR ("INTUBATION TIME")) OR ("DIFFICULTY PERCEPTION")) OR ("PERCEPTION OF DIFFICULTY ")).

## Study selection

The principal investigator (JHT-P) and author CEM-M independently reviewed the titles and abstracts to determine their eligibility. After reviewing the titles and abstracts, the inter-reviewer agreement was 91.2%. Disagreements were resolved through discussion and consensus. When disagreements persisted, a third reviewer (JAR-C) made the final decision.

Following the review of titles and abstracts, full-text articles were assessed for eligibility. JHT-P and CEM-M conducted a full-text review. In addition, a search was conducted for references that included additional studies.

## Data extraction

Information from the articles was extracted by four reviewers (JHT-P, CEM-M, JAR-C, and JHT-C) using an established format with the following variables: lead author, year of publication, geographic location, language, population that was intubated, number of procedures performed during the study, participants who performed intubation, type of device used and number of procedures performed with each device, intubation time, first-attempt success rate, overall success rate, and perceived ease of use of the ETI device. When data were missing or incomplete, attempts were made to contact the study authors to obtain further information, however, we did not receive a response from them.

## Risk of bias in studies

The authors (JHT-P, CEM-M, JAR-C, JHT-C) examined the quality of the CTs. Two reviewers independently assessed each article using the Cochrane risk-of-bias (RoB) tool from the Cochrane Handbook for Systematic Reviews of Interventions. Quality was assessed by evaluating the following potential sources of bias: random sequence generation, allocation sequence concealment, blinding of participants and personnel, blinding of outcome assessment, incomplete outcome data, and selective outcome reporting. Each of the domains judged was classified as follows: high risk of bias, low risk of bias, and some concerns [22].

## Synthesis of information

The studies were described according to the language of the publication, the country where they were conducted, and the participants' educational level. Additionally, the types of devices used and the perceived ease of use were analyzed.

## Results

In June 2024, the database search identified 567 records, with an additional 16 records found through citation searches. After excluding review articles, case reports, editorials, conference abstracts, clinical opinions, and comments, 56 studies were assessed for eligibility, including the 16 from the citation search. Twenty-four reports were excluded because they had different designs, including a single cadaver study, three were excluded because they involved pediatric populations, one was excluded as a duplicate, and one was excluded because it was in Chinese.

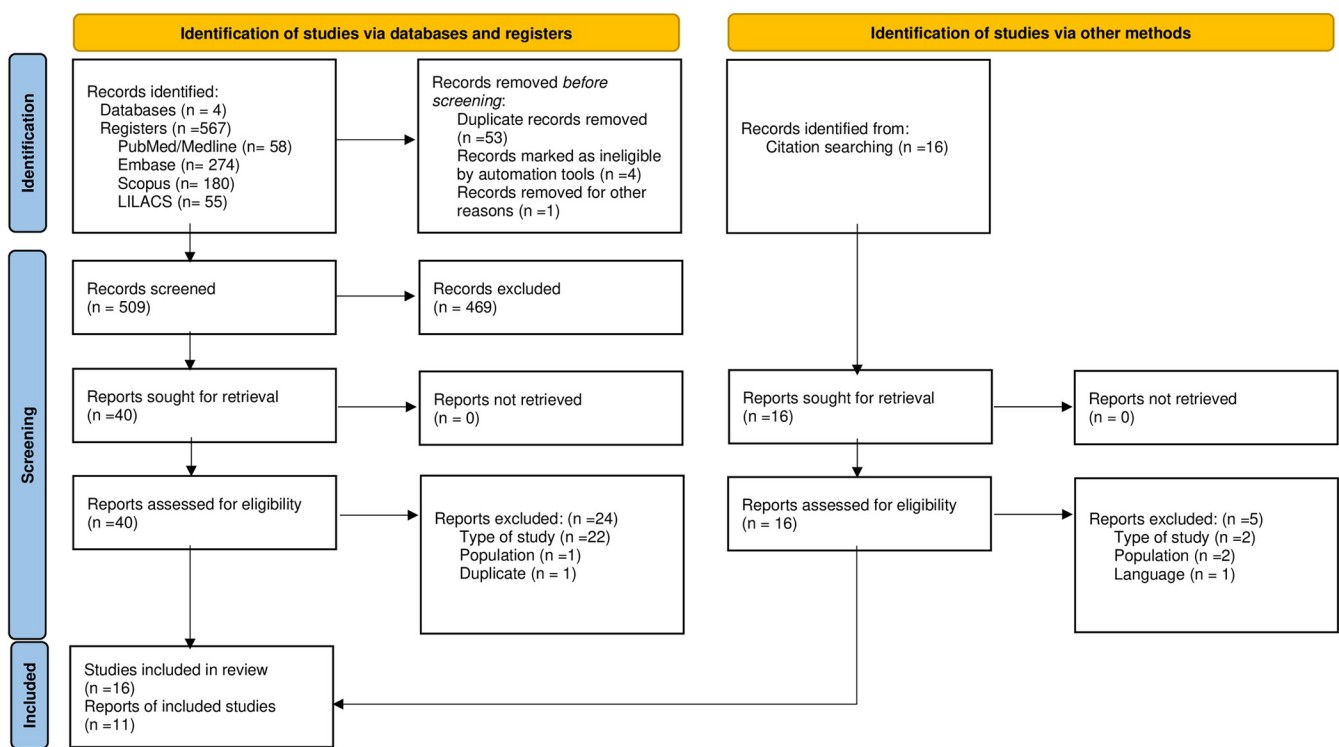

Fig 1. Flow diagram.

*From:* Page MJ, McKenzie JE, Bossuyt PM, Boutron I, Hoffmann TC, Mulrow CD, et al. The PRISMA 2020 statement: an updated guideline for reporting systematic reviews. BMJ 2021;372:n71. doi: 10.1136/bmj.n71. For more information, visit: http://www.prisma-statement.org/

Therefore, 27 CTs were included in the final analysis as outlined in the PRISMA flowchart (Fig 1).

## Characteristics of included studies

The baseline characteristics and description of the CTs that met the inclusion criteria are summarized in Tables 2 and 3. The included studies were published between 2007 and 2022, none were performed in the pre-hospital setting and most were published in English (85.2%) [11, 15, 17–19, 23–40], and only four in Spanish [16, 41–43]. Most studies were prospective randomized [11, 15, 17–19, 23–41, 43], only two were non-randomized [16, 42].

India had the highest number of publications on using devices for blind intubation, with five articles (18.5%) [28, 29, 31, 33, 35]. Poland followed with four articles (14.8%) [19, 24, 27, 40]. Spain (11.1%) [16, 41, 43] and Singapore (11.1%) [15, 32, 39] each had three articles. Germany (7.4%) had two articles [18, 38]. Colombia contributed one study (3.7%) [42] as did Canada [30], Greece [23], Israel [25], Italy [37], Japan [17], Malaysia [34], the United Kingdom [26], and Switzerland [36] each contributed one article (3.7%).

SGDs were used on patients in 51.9% of the studies [11, 15, 28–36, 39, 41, 43], and on simulation manikins in 48.1% [16–19, 23–27, 37, 38, 40, 42].

The population performing SGD intubation consisted primarily of anesthesiologists in 44.4% of the studies [11, 15, 26, 28, 31–33, 35, 36, 39, 41, 43], paramedics in 22.2% [19, 24, 25, 34, 40, 42], medical students in 11.1% [18, 38, 42], nurses in 3.7% [23], physicians in 11.1% [17, 27, 37], residents in 7.4% [16, 30], and inexperienced individuals in 3.7% [37]. One study (3.3%) did not report the population involved [29]. Regarding the participants, most studies

**Table 2. Basic characteristics of the included studies.**

| Author, year (Country) | Intubated Population /n = | *Participants/n = | Procedures |
|---|---|---|---|
| Zuercher, 2022, Swiss | Elective surgery patients/n = 99 | Anesthesiologists/n = 3 | n = 137 |
| De Rosa, 2018, Italy | Manikin/NR | Specialist physician /n = 12 | n = 87 |
| | | Nursing staff/n = 32 | |
| | | Non-medical staff/n = 3 | |
| López Correa, 2016, Spain | Elective surgery patients/n = 80 | Anesthesiologists/n = 3 | n = 119 |
| Ott, 2020, Germany | Manikin/NR | Medical students/n = 126 | n = 629 |
| Fun, 2007, Singapore | Elective surgery patients/n = 60 | Anesthesiologists/NR | n = 180 |
| Gawlowski, 2017, Poland | Manikin/NR | Paramedics/n = 46 | n = 276 |
| Melissopoulou, 2014, Greece | Manikin /NR | Nurses /n = 45 | n = 135 |
| Kurowski, 2015, Poland | Manikin /NR | Emergency Medical Technicians /n = 120 | n = 480 |
| Uribe-Valencia, 2014, Colombia | Manikin /NR | Medical students, army first aid and pre-hospital care personnel /n = 90 | n = 102 |
| Shavit, 2015, Israel | Manikin /NR | Military Doctors/n = 29 | n = 58 |
| Ott, 2015, Germany | Manikin /NR | Medical students/n = 64 | n = 512 |
| Komasawa, 2014, Japan | Manikin /NR | Physicians/n = 20. | NR |
| Michalek, 2010, United Kingdom | Manikin /NR | Physicians/n = 25 | n = 450 |
| Alonso, 2012, Spain | Manikin /NR | Resident doctors /n = 6. | n = 300 |
| Bielski, 2018, Poland | Manikin /NR | Physicians/n = 134 | n = 1206 |
| Szarpak, 2015, Poland | Manikin /NR | Paramedics/n = 210 | n = 1268 |
| Anuradha, 2017, India | Elective surgery patients /n = 80 | Anesthesiologist/n = 1 | n = 101 |
| Darlong, 2011, India | Elective surgery patients /n = 60 | NR | n = 81 |
| Halwagi, 2012, Canada | Elective surgery patients /n = 160 | Anesthesiologist/n = 1 | NR |
| Kapoor, 2014, India | Elective surgery patients /n = 100 | Anesthesiologist/n = 1 | n = 130 |
| Karim, 2011, USA | Elective surgery patients /n = 154 | Anesthesiologists/NR | n = 182 |
| Liu, 2008, Singapore | Elective surgery patients /n = 271 | Anesthesiologists/n = 4 | n = 152 |
| Malhotra, 2016, India | Elective surgery patients /n = 120 | Anesthesiologists/n = 3 | n = 173 |
| Neoh, 2012, Malaysia | Elective surgery patients /n = 160 | Anesthesiologist/n = 1 | n = 232 |
| Teoh, 2007, Singapore | Elective surgery patients /n = 84 | Anesthesiologists/n = 2 | n = 124 |
| Sethi, 2017, India | Elective surgery patients /n = 90 | Anesthesiologist/n = 1 | n = 133 |
| Garzón, 2014, Spain | Elective surgery patients /n = 80 | Anesthesiologists/n = 3 | n = 92 |

n = Number of procedures.

NR = No record.

*Participants = People who perform the intubation procedure.

were conducted with professionals and specialists with some knowledge in the use of SGDs [11, 15–17, 25, 26, 28–36, 39, 41, 43], other studies report participants without experience in SGDs but were subjected to training and instruction prior to the start of the study [18, 19, 23, 24, 27, 37, 38, 40, 42].

## Synthesis of results

A total of 7,339 procedures were performed; two studies did not report the number of procedures [17, 30]. LMA Fastrach was used as the blind intubation device in 81.5% of the studies [11, 15–19, 23, 25, 26, 28–34, 36–39, 41, 43]. The i-gel [17, 23, 26–28, 30, 31, 40] and Air-Q [11, 17, 19, 27, 33–35, 41] were used in 29.6% of the studies. The S.A.L.T. device was used in 11.1% of the studies [19, 24, 42], followed by AuraGain and ILTS used in 11.1% of the studies [17, 18, 35, 36, 38, 43]. The Cobra Perilaryngeal Airway (CobraPLA) was used in 7.4% of the

**Table 3. Characteristics of the included clinical trials.**

| Author, year (Country) | Airway devices comparison | Intubation time (seconds) | First attempt success rate | Overall success rate | Perception of ease |
|---|---|---|---|---|---|
| **Zuercher, 2022, Switzerland** | iLTS-D vs LMA Fastrach | iLTS-D = 43.6 (SD 14) LMA Fastrach = 50.4 (SD 19) | iLTS-D = 42.6% (21/50)[a] LMA Fastrach = 81.6% (40/49)[a] | iLTS-D = 70% (35/50)[a] LMA Fastrach = 91.8% (45/49)[a] | NR |
| **De Rosa, 2018, Italy** | Spritztube vs LMA Fastrach | LMA Fastrach = 79 (IQR 60–90) Spritztube = 90 (IQR 70–110) | NR | LMA Fastrach = 85% (40/47)[b] Spritztube = 100% (47/47)[b] | Ease of insertion LMA Fastrach Very easy = 5 Easy = 32 Moderate = 4 Difficult = 5 Very difficult = 1 Spritztube Very easy = 27 Easy = 18 Moderate = 2 Difficult = 0 Very difficult = 0 |
| **López Correa, 2016, Spain** | AuraGain vs LMA Fastrach | NR | AuraGain = 13.5% (5/37)[a] LMA Fastrach = 66.6% (24/36)[a] | AuraGain = 18.9% (7/37)[a] LMA Fastrach = 73.7% (28/38)[a] | NR |
| **Ott, 2020, Germany** | ILTS-D2 vs LMA Fastrach | First attempt ILTS-D2 = 23 (IQR 20–26) LMA Fastrach = 19 (IQR 17–22) Second attempt ILTS-D2 = 20 (IQR 17–22) LMA Fastrach = 16 (IQR 15–19) Third attempt ILTS-D2 = 18 (IQR 16–21) LMA Fastrach = 15 (IQR 14–18) Fourth attempt ILTS-D2 = 18 (IQR 15–22) LMA Fastrach = 15 (IQR 13–18) Fifth attempt ILTS-D2 = 16 (IQR 15–20) LMA Fastrach = 14 (IQR 13–17) | First attempt ILTS-D2 = 98% (62/63)[b] LMA Fastrach = 100% (63/63)[b] | ILTS-D2 = 97.8% (308/315)**[b] LMA Fastrach = 99.4% (312/314)**[b] | NR |
| **Fun, 2007, Singapore** | LMA Fastrach vs GlideScope | LMA Fastrach = 21 (SD 20.9) GlideScope = 14.3 (SD 17.7) | LMA Fastrach = 63.3% (19/30)[a] GlideScope = 96.6% (29/30)[a] | LMA Fastrach = 90% (27/30)[a] GlideScope = 96.6% (29/30)[a] | NR |
| **Gawlowski, 2017, Poland** | i-gel vs Macintosh Laringoscope | Scenario A i-gel = 12 (IQR 11–13) ML = 19 (IQR 18–21) Scenario B i-gel = 11.5 (IQR 11–13) ML = 20 (IQR 18–22) Scenario C i-gel = 13 (IQR 11–14) ML = 24 (IQR 21–27) | Scenario A[b] i-gel = 95.7% (44/46) ML = 100% (46/46) Scenario B[b] i-gel = 100% (46/46) ML = 84.8% (39/46) Scenario C[b] i-gel = 100% (46/46) ML = 82.6% (38/47) | NR | Ease of use VAS scale 1 (Extremely easy) a 10 (Extremely difficult) Scenario A i-gel = 2 (2–4) ML = 2.5 (2–4) Scenario B i-gel = 2.5 (2–4) ML = 4.5 (3–6) Scenario C i-gel = 2.5 (2–4.5) ML = 6 (4.5–7) |

*(Continued)*

**Table 3.** (Continued)

| Author, year (Country) | Airway devices comparison | Intubation time (seconds) | First attempt success rate | Overall success rate | Perception of ease |
|---|---|---|---|---|---|
| **Melissopoulou, 2014, Greece** | LMA Fastrach vs i-gel | LMA Fastrach(cc) = 21.99 (SD 3.62)<br>i-gel(cc) = 20.00 (SD 3.71)<br>LMA Fastrach(No cc) = 20.60 (SD 3.71)<br>i-gel(No cc) = 18.40 (SD 3.26) | LMA Fastrach(cc) = 77.8%[b]<br>i-gel(cc) = 73.3%[b]<br>LMA Fastrach(No cc) = 86.7%[b]<br>i-gel(No cc) = 82.2%[b] | LMA Fastrach(cc) = 84.4%[b]<br>i-gel(cc) = 82.2%[b]<br>LMA Fastrach(No cc) = 91.9%[b]<br>i-gel(No cc) = 83.7%[b] | NR |
| **Kurowski, 2015, Poland** | S.A.L.T vs Macintosh Laryngoscope | S.A.L.T(cc) = 20.42 (SD 7.54)<br>ML(cc) = 39.43 (SD 8.23)<br>S.A.L.T(No cc) = 17.97 (SD 5.33)<br>ML(No cc) = 31.52 (SD 7.23) | S.A.L.T(cc) = 82%[b]<br>ML(cc) = 65.42%[b]<br>S.A.L.T(No cc) = 91%[b]<br>ML(No cc) = 76.54%[b] | S.A.L.T(cc) = 84.33%[b]<br>ML(cc) = 71.23%[b]<br>S.A.L.T(No cc) = 96.23%[b]<br>ML(No cc) = 82.45%)[b] | NR |
| **Uribe-Valencia, 2014, Colombia** | S.A.L.T vs Not reported | S.A.L.T first attempt = 16 (IQR = 13–20)<br>S.A.L.T second attempt = 19 (IQR = 18–25,75)<br>S.A.L.T thirth attempt = 24 (IQR = NR) | S.A.L.T = 90% (81/90)[b] | S.A.L.T = 98.8% (89/90)[b] | Difficulty of procedure S.A.L.T =<br>Very easy = 48.9% (44/90)<br>Easy = 47.8% (43/90)<br>No easy nor difficult = 2.2% (2/90)<br>Difficult = 0% (0/90)<br>Very difficult = 1.1% (1/90) |
| **Shavit, 2015, Israel** | LMA Fastrach vs Direct laryngoscope | LMA Fastrach = 54.2 (SD 9.8)<br>DL = 42.8 (SD 9.7) | LMA Fastrach = 93.1% (27/29)[b]<br>DL = 62% (18/29)[b] | NR | Ease of use<br>Five-point Likert Scale<br>1 = Strongly disagree<br>2 = Disagree.<br>3 = Neither agree nor disagree<br>4 = Agree<br>5 = Strongly agree<br>LMA Fastrach = 4 (IQR 4–5)<br>DL = 3 (IQR 2–4) |
| **Ott, 2015, Germany** | iLTS-D vs LMA Fastrach | Trial 1<br>iLTS-D = 19 (IQR 16.0–22.75)<br>LMA Fastrach = 19 (IQR 16.0–24.75)<br>Trial 2<br>iLTS-D = 16 (IQR 14.0–19.0)<br>LMA Fastrach = 15 (IQR 12.25–17.0)<br>Trial 3<br>iLTS-D = 15 (IQR 13.0–17.75)<br>LMA Fastrach = 14 (IQR 12.0–16.0)<br>Trial 4<br>iLTS-D = 14 (IQR 12.0–17.0)<br>LMA Fastrach = 14 (IQR 12.0–16.5) | Trial 1[b]<br>iLTS-D = 98.4% (63/64)<br>LMA Fastrach = 98.4% (63/64) | iLTS-D = 98.8% (253/256) **[b]<br>LMA Fastrach = 99.6% (255/256)**[b] | Device qualification<br>1 (best) a 6 (worst)<br>iLTS-D = 1.5 (IQR 1–2)<br>LMA Fastrach = 2 (IQR 1–2) |
| **Komasawa, 2014, Japan** | Air-Q vs aura-i vs i-gel vs LMA Fastrach | NR | NR | Air-Q = 78.94% (15/19)[b]<br>aura-i = 73.68% (14/19)[b]<br>i-gel = 75% (12/16)[b]<br>LMA Fastrach = 55.55% (10/18)[b] | NR |

*(Continued)*

**Table 3.** (Continued)

| Author, year (Country) | Airway devices comparison | Intubation time (seconds) | First attempt success rate | Overall success rate | Perception of ease |
|---|---|---|---|---|---|
| **Michalek, 2010, United Kingdom** | LMA Fastrach vs i-gel | NR | NR | LMA Fastrach = 88.4% (199/225)[b] <br> i-gel = 50.7% (114/225)[b] | NR |
| **Alonso, 2012, Spain** | LMA Fastrach vs NR | LMA Fastrach = 25 (SD 11.71) | LMA Fastrach = 84% (252/300)[b] | LMA Fastrach = 97.3% (292/300)[b] | NR |
| **Bielski, 2018, Poland** | i-gel vs Air-Q vs Macintosh Laryngoscope | i-gel(No cc) = 19 (IQR 16–25) <br> Air-Q(No cc) = 23 (IQR 16.5–31) <br> ML(No cc) = 24.5 (IQR 21–32) <br> i-gel(cc) = 18.5 (IQR 17–27) <br> Air-Q(cc) = 20.5 (IQR 18–32) <br> ML(cc) = 41.5 (IQR 35–49) <br> i-gel(VADcc) = 19.5 (IQR 17.5–27.5) <br> Air-Q(VADcc) = 22 (IQR 19.5–35) <br> ML(VADcc) = 49.5 (IQR 44–67) | i-gel(No cc) = 81.3% (109/134)[b] <br> Air-Q(No cc) = 74.6% (100/134)[b] <br> ML(No cc) = 71.7% (96/134)[b] <br> i-gel(cc) = 79.9% (107/134)[b] <br> Air-Q(cc) = 74.6% (100/134)[b] <br> ML(cc) = 41.8% (56/134)[b] <br> i-gel(VADcc) = 73.9% (99/134)[b] <br> Air-Q(VADcc) = 64.9% (87/134)[b] <br> ML(VADcc) = 23.1% (31/134)[b] | i-gel(No cc) = 100% (134/134)[b] <br> Air-Q(No cc) = 98.5% (132/134)[b] <br> ML(No cc) = 94.8% (127/134)[b] <br> i-gel(cc) = 100% (134/134)[b] <br> Air-Q(cc) = 74.6% (100/134)[b] <br> ML(cc) = 41.8% (56/134)[b] <br> i-gel(VADcc) = 94.8% (127/134)[b] <br> Air-Q(VADcc) = 85.8% (115/134)[b] <br> ML(VADcc) = 31.3% (42/134)[b] | Ease of use (1–100 Scale) <br> i-gel(No cc) = 35 (IQR 27–37) <br> Air-Q(No cc) = 36 (IQR 27.5–40) <br> ML(No cc) = 42 (IQR 36–47) <br> i-gel(cc) = 36 (IQR 26–38) <br> Air-Q(cc) = 38 (IQR 27–44) <br> ML(cc) = 69 (IQR 51–77) <br> i-gel(VADcc) = 36 (IQR 27–39) <br> Air-Q(VADcc) = 40 (IQR 29–46) <br> ML(VADcc) = 80 (IQR 63–84) |
| **Szarpak, 2015, Poland** | LMA Fastrach vs CobraPLA vs S.A.L.T vs air-Q | LMA Fastrach = 39.9 <br> CobraPLA = 32.5 <br> S.A.L.T = 17.2 (SD 4.6) <br> air-Q = 49.2 (SD 19.6) | LMA Fastrach = 76.7%[b] <br> CobraPLA = 58.1%[b] <br> S.A.L.T = 90.5%[b] <br> air-Q = 54.7%[b] | LMA Fastrach = 86.7%[b] <br> CobraPLA = 85.7%[b] <br> S.A.L.T = 100%[b] <br> air-Q = 71.4%[b] | Ease of use <br> 1 (Very easy) a 5 (Very difficult) <br> Mean = <br> LMA Fastrach = 2.2 <br> CobraPLA = 2.4 <br> S.A.L.T = 1.6 <br> air-Q = 3.3 |
| **Anuradha, 2017, India** | i-gel vs LMA Fastrach | i-gel = 15.88 (SD 2.49) <br> LMA Fastrach = 16.31 (SD 3.04) | i-gel = 60% (24/40)[a] <br> LMA Fastrach = 87.5% (35/40)[a] | I-gel = 72.5% (29/40)[a] <br> LMA Fastrach = 95% (38/40)[a] | NR |
| **Darlong, 2011, India** | CobraPLA vs LMA Fastrach | CobraPLA = 9 (IQR 3–106) <br> LMA Fastrach = 4 (IQR 3–90) | CobraPLA = 63.33% (19/30)[a] <br> LMA Fastrach = 70% (21/30)[a] | CobraPLA = 86.7% (26/30)[a] <br> LMA Fastrach = 90% (27/30)[a] | NR |
| **Halwagi, 2012, Canada** | i-gel vs LMA Fastrach | i-gel = 22 (SD 13) <br> LMA Fastrach = 30 (SD 31) | i-gel = 69% (55/80)[a] <br> LMA Fastrach = 74% (59/80)[a] | I-gel = 73% (58/80)[a] <br> LMA Fastrach = 91% (73/80)[a] | NR |
| **Kapoor, 2014, India** | i-gel vs LMA Fastrach | i-gel = 24.04 (SD 9.42) <br> LMA Fastrach = 20.96 (SD 8.23) | i-gel = 66% (33/50)[a] <br> LMA Fastrach = 74% (37/50)[a] | i-gel = 82% (41/50)[a] <br> LMA Fastrach = 96% (48/50)[a] | NR |
| **Karim, 2011, USA** | LMA Fastrach vs Air-Q | LMA Fastrach = 27 (IQR 23–32) <br> Air-Q = 35 (IQR 25–155) | LMA Fastrach = 94% (71/76)[a] <br> Air-Q = 70% (54/78)[a] | LMA Fastrach = 99% (75/76)[a] <br> Air-Q = 77% (60/78)[a] | NR |
| **Liu, 2008, Singapore** | LMA Fastrach vs LMA CTrach | LMA Fastrach = 100 (IQR 74–121) <br> LMA CTrach = 116 (IQR 82–156) | LMA Fastrach = 67.9% (93/137)[a] <br> LMA CTrach = 93.3% (125/134)[a] | LMA Fastrach = 96.4% (132/137)[a] <br> LMA CTrach = 100% (134/134)[a] | NR |

*(Continued)*

**Table 3.** (*Continued*)

| Author, year (Country) | Airway devices comparison | Intubation time (seconds) | First attempt success rate | Overall success rate | Perception of ease |
|---|---|---|---|---|---|
| **Malhotra, 2016, India** | LMA Fastrach vs Air- Q | LMA Fastrach = 20.15 (SD 15) Air-Q = 15 (SD 5) | LMA Fastrach = 81.8% (45/60)[a] Air-Q = 67.2% (39/60)[a] | LMA Fastrach = 91.6% (55/60)[a] Air-Q = 96.6% (58/60)[a] | Ease of Intubation = Easy Moderate Difficult LMA Fastrach Easy = 85.4% (47/55) Moderate = 7.2% (4/55) Difficult = 7.2% (4/55) Air-Q Easy = 87.9% (51/58) Moderate = 12% (7/58) Difficult = 0 |
| **Neoh, 2012, Malaysia** | Air-Q vs LMA Fastrach | NR | Air-Q = 63.75% (51/80)[a] LMA Fastrach = 82.3% (65/79)[a] | Air-Q = 75% (60/80)[a] LMA Fastrach = 97.4% (77/79)[a] | Ease of Intubación = Easy/Moderate/Difficult. Air-Q Easy = 75% (60/80) Moderate = 0 Difficult = 25% (20/80) LMA Fastrach Easy = 93.67% (74/79) Moderate = 3.79% (3/79) Difficult = 2.53% (2/79) |
| **Teoh, 2007, Singapore** | Single use LMA Fastrach vs Reusable LMA Fastrach | Single use LMA Fastrach = 101.4 (SD 63.2) Reusable LMA Fastrach = 90.4 (SD 46.1) | Single use LMA Fastrach = 62.5% (25/40)[a] Reusable LMA Fastrach 67.5% (27/40)[a] | Single use LMA Fastrach = 100% (40/40)[a] Reusable LMA Fastrach = 100% (40/40)[a] | Ease of use VAS 0 (very easy) a 10 (very difficult) Single use LMA Fastrach = 10 (IQR 0–100) Reusable LMA Fastrach = 10 (IQR 0–100) |
| **Sethi, 2017, India** | AuraGain vs Air-Q | AuraGain = 26 (IQR 25–27) Air-Q = 22 (IQR 21–24) | AuraGain = 35,6% (16/45)[a] Air-Q = 68.9% (31/45)[a] | AuraGain = 53.3% (24/45)[a] Air-Q = 80% (36/45)[a] | Ease of Intubación = Easy Moderate Difficult Imposible AuraGain Easy = 22.2% (10/45) Moderate = 17.7% (8/45) Difficult = 13.3% (6/45) Imposible = 46.6% (21/45) Air-Q Easy = 42.2% (19/45) Moderate = 24.4% (11/45) Difficult = 13.3% (6/45) Imposible = 20% (9/45) |

**Table 3.** (Continued)

| Author, year (Country) | Airway devices comparison | Intubation time (seconds) | First attempt success rate | Overall success rate | Perception of ease |
|---|---|---|---|---|---|
| **Garzón, 2014, Spain** | LMA Fastrach vs Air-Q | NR | LMA Fastrach = 66.67% (24/36)[a] Air-Q = 58.33% (14/24)[a] | LMA Fastrach = 78.95% (30/38)[a] Air-Q = 75% (24/32)[a] | NR |

iLTS-D = intubating laryngeal tube; ILTS-D2 = intubating laryngeal tube 2; LMA Fastrach = Fastrach intubating laryngeal mask; ML = Macintosh Laryngoscope

S.A.L.T = Supraglottic airway laryngopharyngeal tube; LMA Fastrach(cc) = Fastrach intubating laryngeal mask with chest compressions; i-gel(cc) = i-gel with chest compressions

LMA Fastrach(No cc) = LMA Fastrach without chest compressions; i-gel(No cc) = i-gel without chest compressions; S.A.L.T(cc) = S.A.L.T with chest compressions

S.A.L.T(No cc) = S.A.L.T without chest compressions; DL = Direct laringoscope; VADcc = Difficult airway with continuous chest compression; VAS = Visual Analogue Scale

SD = Standard Deviation.

IQR: Interquartile range.

NR = No record

** = Calculated data

[a] = Humans

[b] = Manikin

studies [19, 29], while the Spritztube [37], and single-use LMA Fastrach [15] were each used in only one study.

Of the included studies, 16 articles did not report the method used to assess the ease of use of the SGDs [11, 16, 17, 23, 24, 26, 28–32, 36, 38, 39, 41, 43]. There was significant heterogeneity in the assessment methods for perceived ease of use of the SGD. Studies that used a 1-to-5 scale for assessing perceived ease of use reported a rating of "very easy" to "easy" for the S.A.L. T. device [19, 42], "easy" to "moderate" for the LMA Fastrach [19, 37] and CobraPLA [19], and "moderate" to "difficult" for the Air-Q [19].

One study used the 0–10 visual analog scale (VAS), i-gel was a device easy to use [40]. Conversely, the classic and single-use versions of LMA Fastrach were found to be very difficult to use [15]. The only study using the Likert scale reported that participants agreed it was easy to intubate with LMA Fastrach [25].

**a. Blind intubation time with SGD.** Blind intubation time was an outcome that was evaluated in different ways in the studies. Of the 27 articles included in the study, five (18.5%) did not report the time required for intubation [17, 26, 34, 41, 43]. In studies conducted on manikins, the S.A.L.T device had the shortest intubation times, ranging from 16 seconds [42] to 20.42 seconds [24]. The i-gel, in a study by Gawlowski et al., had an intubation time of 11.5 seconds [40]. Among the studies involving patients, LMA Fastrach had intubation times that ranged from 4 seconds [29] to 100 seconds [32] showing high variability, this was due to the way intubation times were evaluated and i-gel had intubation times between 15.88 seconds [28] and 24.04 seconds [31]. Both devices demonstrated the shortest intubation times [11, 15, 28–33, 36, 39] (Table 3).

**b. First-attempt success rate.** Three articles did not report the first attempt success rate [17, 26, 37]. The device with the lowest success rate on the first attempt was AuraGain (13.5%) [43], followed by CobraPLA, with success rates ranging from 58.1% to 63.3% [29]. Two studies reported a 100% successful rate on the first attempt using LMA Fastrach and i-gel [38, 40]. The most frequently compared device in the CTs was the LMA Fastrach, followed by the i-gel and Air-Q.

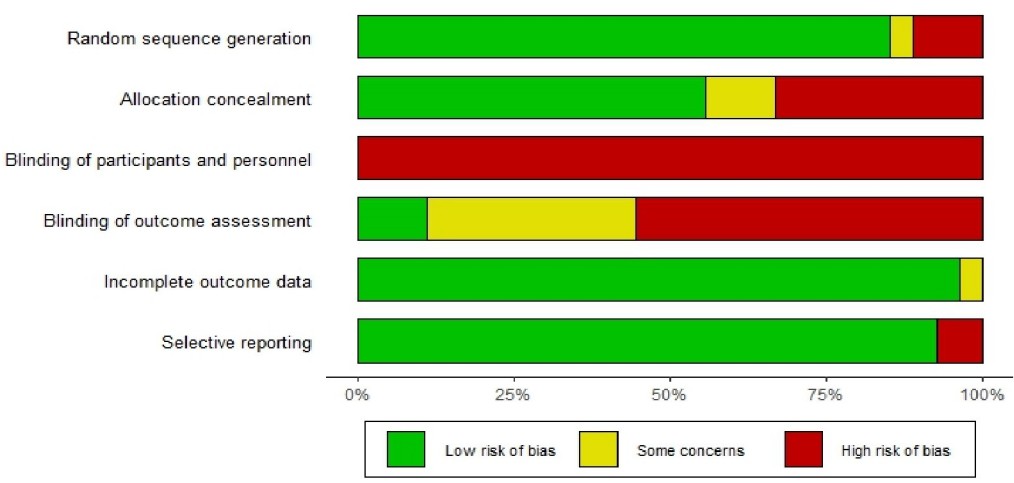

**Fig 2. Risk bias of included studies.** "some concerns" are due to lack of detailed reports.

Regarding LMA Fastrach, three studies reported a success rate greater than 90% [11, 18, 38], while three studies reported a success rate of 66% [15, 41, 43]. S.A.L.T was reported in three studies with success rates ranging from 82% in scenarios with chest compressions [24] to 91% in scenarios without chest compressions [24, 42] (Table 3).

**c. Overall success rate.** Two articles did not report the overall success rate [25, 40]. A 100% success rate was achieved with i-gel, S.A.L.T., LMA Fastrach, and single-use LMA Fastrach [15, 19, 27]. However, it should be noted that in other studies, the modified S.A.L.T., Air-Q, ILTS-D, iLTS-D2, and the devices above mentioned achieved intubation success rates greater than 90% [11, 16, 18, 23, 24, 27–34, 36, 38, 39, 42]. The lowest overall success rate for blind ETI was observed with AuraGain (18.9%) [43] (Table 3).

## Risk of bias in the study

Of the 27 studies, 23 clearly reported random sequence generation methods [11, 15, 17–19, 23–36, 38–41], and 15 described allocation sequence concealment methods [11, 15, 23, 28–36, 39, 41, 43]. None of the studies blinded participants or personnel. Only three studies reported methods for outcome assessment blinding [16, 33, 34]. All CTs were classified with a low risk of bias for the domains "incomplete outcome data" and "selective outcome reporting," except for one CT [28], which was classified as having some concerns and a high risk of bias in these two domains, respectively. Details of the risk of bias assessment are presented in Fig 2.

## Discussion

According to the results of our study, the LMA Fastrach is the most studied intubation SGD to date. The difficulties encountered during "blind" insertion have led to the recommendation of using the 90° rotational technique, the use of a lightwand or combined videolaryngoscopy-bougie techniques, which may suggest a tendency to discourage interest in intubation with SGDs [44]. However, all recent guidelines for difficult airway management recommend the use of SGDs as a rescue measure for oxygenation when other resources fail, and even suggest that they should be present on portable difficult airway carts, but the recommendation of which SGDs to use remains unclear [45].

This systematic review summarizes the effectiveness of SGDs in blind ETI in different scenarios, encompassing human subjects and simulation manikins. Additionally, it highlights the

wide geographical distribution of the studies, which supports universally representative results.

Differences in intubation time were observed when comparing different types of devices. These disparities may be due to heterogeneity in the definitions, forms and methods in which intubation time is assessed, variations in intubation scenarios and the participants' level of training and experience in SGD management.

Studies demonstrated S.A.L.T. as the device with the shortest intubation times. However, since these studies used simulation manikins, the actual intubation time in real-world clinical scenarios might differ [24, 42]. Anand et al. suggest that the insertion time of SGDs in humans could be prolonged due to possible anatomical differences between patients and simulation models, varying degrees of muscle relaxation, and the need to overcome a learning curve [46]. Darlong et al. found that the LMA Fastrach facilitated the passage of the orotracheal tube more swiftly than other devices, possibly due to its rigid handle and body, which stabilize the device and ease the passage of the orotracheal tube into the airway without deviation [29]. Conversely, the single-use LMA Fastrach, Aura-Gain, and Air-Q appeared to take longer to achieve blind ETI. Intubation time with these devices may be longer because, while they can ventilate the patient, they can also move and deflect the tube away from the midline, thereby delaying the passage of the tube through the vocal cords and increasing the risk of airway injury [29, 35].

The LMA Fastrach and i-gel exhibited the highest first-attempt success rates, followed by iLTS-D2 and S.A.L.T., all surpassing an 80% success rate. Darlong et al., in their comparison of CobraPLA to LMA Fastrach, observed a higher blind ETI success rate with the latter, although without statistical significance. Through pilot testing, they identified the potential resistance mechanism of the CobraPLA and directed the PVC tube down the midline without advancing into the perilaryngeal space or esophagus or injuring the airway. The CobraPLA has certain advantages, including ease of insertion and good airway sealing, which reduces overall insertion time. In addition, the CobraPLA does not require special tubing and does not need to be removed after intubation, unlike the LMA Fastrach, whose weight can exert pressure on the hypopharynx even when the cuff is deflated [29].

While it is crucial for the healthcare provider to establish a definitive airway rapidly, maintaining adequate patient ventilation is equally important regardless of the attempts made. Consequently, the overall success rates become paramount.

This systematic review revealed that, except for the LMA Fastrach, which was the first device designed for blind ETI, most devices exhibited comparable success rates. CobraPLA, iLTS, and i-gel performed similarly, boasting features that facilitate airway management, softer components, and the convenience of not having to remove the device once intubation is achieved. Ahn et al. noted that the devices with the best overall success rates in unassisted airway management included LMA-CTrach, single-use LMA Fastrach, and Air-Q [20]. Teoh and Lim found no differences in overall success rates or ease of insertion between single-use LMA Fastrach and LMA Fastrach; however, the single-use LMA Fastrach offers some additional advantages, such as the option to use PVC tubes and a reduced risk of disease transmission [15].

The systematic review produced some relevant findings, although we acknowledge several limitations. Firstly, the studies analyzed were methodologically and clinically diverse. Participants performing intubations were from different disciplines, and interventions occurred in diverse settings, encompassing procedures on patients, simulation manikins, and cadavers. Secondly, despite a thorough search, the possibility of overlooking relevant studies, particularly within gray literature, remains. Thirdly, the studies were not blinded to the investigators inserting the tracheal tube using the SGD. Due to the methodological heterogeneity across studies, conducting a meta-analysis to determine the device with the best performance proved

unfeasible, constituting another limitation. Lastly, the included CTs had a high risk of bias or some concerns in the assessment of allocation sequence concealment and blinding of outcome assessors, as evaluated using the Cochrane risk-of-bias (RoB) tool.

Despite the limitations described above, this systematic review represents the first attempt to comprehensively investigate the effectiveness of different SGDs for blind ETI in different populations, including humans and simulation manikins. The study evaluates multiple SGDs for blind intubation, assessing them based on intubation time, first-attempt intubation rate, and overall success rate.

## Conclusion

This review demonstrates that the new SGDs have not surpassed the LMA Fastrach for blind ETI regarding first-attempt and overall success rates.

The single-use LMA Fastrach combines the efficacy of the reusable LMA Fastrach with the features of other SGDs, potentially making it a suitable replacement for the LMA Fastrach for blind ETI.

I-gel emerges as a good alternative among SGDs for blind ETI, showing good results in terms of first-attempt success and overall success rates. In contrast, the AuraGain showed the poorest performance for ETI. Although it ventilated the patients, its use for blind ETI may not be recommended.

## Supporting information

**S1 File. PRISMA checklist.**
(DOCX)

**S2 File. PRISMA abstracts checklist.**
(DOCX)

**S1 Table. Dataset with the data of the included studies.**
(XLSX)

**S2 Table. Risk of bias in the included studies.**
(DOCX)

## Author Contributions

**Conceptualization:** Jorman H. Tejada-Perdomo, Carlos E. Melo-Moreno, Jorge A. Ramos-Castaneda.

**Data curation:** Jorman H. Tejada-Perdomo, Carlos E. Melo-Moreno, Jorge A. Ramos-Castaneda.

**Formal analysis:** Jorman H. Tejada-Perdomo, Carlos E. Melo-Moreno, Jesús H. Tovar-Cardozo, Jorge A. Ramos-Castaneda.

**Funding acquisition:** Jorman H. Tejada-Perdomo, Jesús H. Tovar-Cardozo.

**Investigation:** Jorman H. Tejada-Perdomo, Carlos E. Melo-Moreno.

**Methodology:** Jorman H. Tejada-Perdomo, Carlos E. Melo-Moreno, Jorge A. Ramos-Castaneda.

**Writing – original draft:** Jorman H. Tejada-Perdomo, Carlos E. Melo-Moreno.

**Writing – review & editing:** Jorman H. Tejada-Perdomo, Carlos E. Melo-Moreno, Jorge A. Ramos-Castaneda.

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
