## [Decision Letter · Decision Letter 0]

27 Sep 2024

PONE-D-24-29958Supraglottic airway devices for blind endotracheal intubation: A systematic review.PLOS ONE

Dear Dr. Tejada,

Thank you for submitting your manuscript to PLOS ONE. After careful consideration, we feel that it has merit but does not fully meet PLOS ONE’s publication criteria as it currently stands. Therefore, we invite you to submit a revised version of the manuscript that addresses the points raised during the review process. Please submit your revised manuscript by Nov 11 2024 11:59PM.If you will need more time than this to complete your revisions, please reply to this message or contact the journal office at plosone@plos.org. Please include the following items when submitting your revised manuscript:A rebuttal letter that responds to each point raised by the academic editor and reviewer(s). You should upload this letter as a separate file labeled 'Response to Reviewers'.A marked-up copy of your manuscript that highlights changes made to the original version. You should upload this as a separate file labeled 'Revised Manuscript with Track Changes'.An unmarked version of your revised paper without tracked changes. You should upload this as a separate file labeled 'Manuscript'.

We look forward to receiving your revised manuscript.

Kind regards,

Silvia Fiorelli

Academic Editor

PLOS ONE

Journal Requirements:

3. Thank you for stating the following financial disclosure: “Publication was funded by Universidad Surcolombiana.”

4. We note that your Data Availability Statement is currently as follows: “All relevant data are within the manuscript and in Supporting Information files.”

Please confirm at this time whether or not your submission contains all raw data required to replicate the results of your study. Authors must share the “minimal data set” for their submission. PLOS defines the minimal data set to consist of the data required to replicate all study findings reported in the article, as well as related metadata and methods (https://journals.plos.org/plosone/s/data-availability#loc-minimal-data-set-definition). For example, authors should submit the following data: - The values behind the means, standard deviations and other measures reported; - The values used to build graphs; - The points extracted from images for analysis. Authors do not need to submit their entire data set if only a portion of the data was used in the reported study. If your submission does not contain these data, please either upload them as Supporting Information files or deposit them to a stable, public repository and provide us with the relevant URLs, DOIs, or accession numbers. For a list of recommended repositories, please see https://journals.plos.org/plosone/s/recommended-repositories. If there are ethical or legal restrictions on sharing a de-identified data set, please explain them in detail (e.g., data contain potentially sensitive information, data are owned by a third-party organization, etc.) and who has imposed them (e.g., an ethics committee). Please also provide contact information for a data access committee, ethics committee, or other institutional body to which data requests may be sent. If data are owned by a third party, please indicate how others may request data access.

Additional Editor Comments:

please carefully assess all the reviewers comments

Reviewers' comments:

Reviewer's Responses to Questions

**Comments to the Author**

1. Is the manuscript technically sound, and do the data support the conclusions?

Reviewer #1: Partly

Reviewer #2: Yes

Reviewer #3: Yes

2. Has the statistical analysis been performed appropriately and rigorously? 

Reviewer #1: N/A

Reviewer #2: Yes

Reviewer #3: Yes

3. Have the authors made all data underlying the findings in their manuscript fully available?

Reviewer #1: Yes

Reviewer #2: Yes

Reviewer #3: Yes

4. Is the manuscript presented in an intelligible fashion and written in standard English?

Reviewer #1: Yes

Reviewer #2: Yes

Reviewer #3: Yes

5. Review Comments to the Author

Reviewer #1: I have stated all my comments in the file attached. This text is just to fill the minimum required characters of 200 in this field, which is not helpful and should be changed to avoid extra (unpaid) work for the reviewers.

Reviewer #2: I think this is an important topic which will be very relevant for clinical application. I commend the authors for the hard work they have put into this manuscript. I appreciate the time taken to revise this manuscript from the initial submission to this current submission as a systematic review. I think it is well written, addresses the primary and secondary outcomes

Reviewer #3: The topic of using supraglottic airway devices remains relevant despite the increasing use of video laryngoscopes, which do not provide an absolute solution for all patients. For safety reasons, skills in emergency intubation with supraglottic devices should be maintained, and it is important to compare the available devices to choose the most suitable one. The presented study is interesting and well-written. However, the data is heterogeneous, including studies on patients, mannequins, and cadavers, which creates some confusion and makes a meta-analysis difficult.

The feedback from previous reviewers has been well incorporated into the new text. However, the inclusion of a single study involving cadavers may introduce more bias than benefit to the analysis and could be excluded to limit the focus to studies involving patients and mannequins, which already exhibit some heterogeneity.

Regarding the populations performing supraglottic device intubation, it appears they were individuals already trained in using the device rather than participants in a training study. This clarification would be valuable for interpreting the results.

The distribution of devices used seems well represented across continents, with no clustering of studies on a single model in any country or continent. This could also be mentioned in the discussion to reinforce the universality of the message.

6. PLOS authors have the option to publish the peer review history of their article (what does this mean?). If published, this will include your full peer review and any attached files.

Reviewer #1: **Yes: **Stephan Katzenschlager

Reviewer #2: No

Reviewer #3: No

---

## [Author Response · Author response to Decision Letter 0]

30 Oct 2024

Neiva October 29, 2024

Dra.

Silvia Fiorelli

Academic Editor

Journal PLOS ONE

Dear Dra Silvia,

Thank you very much for reviewing our manuscript titled “Supraglottic airway devices for blind endotracheal intubation: A systematic review” in PLOS ONE. We appreciate the comments made by each of the reviewers and we have responded point by point to each of the questions raised by them. In our opinion, the quality of the manuscript has increased. 

We have attached a tracked version of the manuscript so you can see how and where we have modified the original manuscript.

Looking forward to hearing from you,

Sincerely,

Jorman Harvey Tejada Perdomo 

Corresponding Author 

Journal Requirements:

- Please ensure that your manuscript meets PLOS ONE's style requirements, including those for file naming. The PLOS ONE style templates can be found at https://journals.plos.org/plosone/s/file?id=wjVg/PLOSOne_formatting_sample_main_body.pdfand
https://journals.plos.org/plosone/s/file?id=ba62/PLOSOne_formatting_sample_title_authors_affiliations.pdf

Thank you very much for the clarification. We took into account the editorial standards of the journal.

- Please note that funding information should not appear in any section or other areas of your manuscript. We will only publish funding information present in the Funding Statement section of the online submission form. Please remove any funding-related text from the manuscript.

Thank you for your valuable comment. We removed funding-related text from the manuscript.

- Thank you for stating the following financial disclosure: “Publication was funded by Universidad Surcolombiana.” Please state what role the funders took in the study. If the funders had no role, please state: "The funders had no role in study design, data collection and analysis, decision to publish, or preparation of the manuscript." If this statement is not correct you must amend it as needed. Please include this amended Role of Funder statement in your cover letter; we will change the online submission form on your behalf.

Following up on your comment about the role of the funder, the funding statement was included in the cover letter that is attached again in this version. Thank you very much.

- We note that your Data Availability Statement is currently as follows: “All relevant data are within the manuscript and in Supporting Information files.”

Please confirm at this time whether or not your submission contains all raw data required to replicate the results of your study. Authors must share the “minimal data set” for their submission. PLOS defines the minimal data set to consist of the data required to replicate all study findings reported in the article, as well as related metadata and methods (https://journals.plos.org/plosone/s/data-availability#loc-minimal-data-set-definition). For example, authors should submit the following data: - The values behind the means, standard deviations and other measures reported; - The values used to build graphs; - The points extracted from images for analysis. Authors do not need to submit their entire data set if only a portion of the data was used in the reported study. If your submission does not contain these data, please either upload them as Supporting Information files or deposit them to a stable, public repository and provide us with the relevant URLs, DOIs, or accession numbers. For a list of recommended repositories, please see https://journals.plos.org/plosone/s/recommended-repositories. If there are ethical or legal restrictions on sharing a de-identified data set, please explain them in detail (e.g., data contain potentially sensitive information, data are owned by a third-party organization, etc.) and who has imposed them (e.g., an ethics committee). Please also provide contact information for a data access committee, ethics committee, or other institutional body to which data requests may be sent. If data are owned by a third party, please indicate how others may request data access.

Thank you for your comment. Regarding the data availability statement, this point is also added to the document and all available data are attached as supplementary information files.

Reviewer #1: Comments

Methods

- Comment #1: Have you registered this SR at Prospero? Please state the PROSPERO registration ID in the method section.

Thanks for your comment. This systematic review was not registered in PROSPERO; however, we will take this comment into account for our future work. 

- Comment #2: The search terms used are focused on prehospital emergency medicine. However, in the eligibility criteria, it is stated that “The studies involved patients under general anesthesia for surgery, simulation manikins, or cadavers.” Were no studies found in the prehospital setting? This would result from the search and not an inclusion or exclusion criteria. In-/Exclusion criteria should be defined upfront and, in this case include the prehospital setting. Please revise this issue; if no prehospital studies were found, briefly state this in the results.

Thanks for your comment. The search was not limited to studies in prehospital emergency medicine. Some studies in the prehospital setting were identified, however, they were not controlled trials and were therefore not included in the systematic review. With this comment in mind, we have added it in the results.

- Comment #3: Please indicate if a professional librarian performed the search and defined the search strategy.

The search strategy was built with the support of an expert in scientific literature search. And the search was conducted by researchers who have experience in scientific literature review.

We added this information in the methodology section.

- Comment #4: Exclusion of pediatric studies: What is the reason for excluding these studies? Airway management is even more time-sensitive in children compared to adults. Further, hypoxia is the leading reversible/preventable cause of cardiac arrest. Since you have identified pediatric studies, I recommend including those in your systematic review and reporting them as a subgroup. This would enhance the quality of your work.

We appreciate your comment. We understand and share the reviewer's opinion regarding airway management in children. During the literature search we identified a systematic review of exclusively pediatric population (https://doi.org/10.1038/s41598-023-45173-5), so in this systematic review we focused on supraglottic devices for the adult population.

- Comment #5: Contact of the study authors: This is valuable information and a great effort from the team to (try to) contact the study team. How many responses did you get? If none, you can also state this. 

We contacted some authors and did not receive a response from them. Following this comment, we added in methodology that “we did not receive a response from them”

Results 

- Comment #1: Please comment on how many studies were prospective, retrospective, or randomized controlled trials. This information can also be included in Table 2 or 3. Another reviewer previously mentioned this.

We appreciate your evaluation. Following your suggestions, we have added information on the methodological design of the included studies to the results.

- Comment #2: It is crucial that you stated who performed the SGD intubation. By “anesthesiologists,” do you mean that the studies were conducted in the OR? For example, Anesthesiologists work as prehospital physicians in Germany due to the lack of a dedicated emergency medicine specialty. I recommend reporting the specialty (anesthesiology, internal medicine, emergency medicine…) as stated in the study or the level of training (resident, consultant). 

Thanks for your comment. In our review, participants refer to the people who performed the intubation procedure. All studies that had anesthesiologists as participants were conducted in elective surgery patients. We have added a comment in Table 2 clarifying the definition of participants.

- Comment #3: You have included studies in humans, mannequins, and cadavers. Please report if the success rates were on humans, mannequins or cadavers.

Following your comment, in Table 3 we have indicated the type of intubated population. In this same table you can see the results by outcome. Thank you very much.

- Comment #4: Among the studies involving patients, LMA Fastrach and i-gel demonstrated the shortest intubation times (11,19,24,25,27,28,36,37,40,41).” Please state the intubation times.

Thank you very much for the accuracy. We have added this information to the results.

- Comment #5: You have extracted data on comparing SGA vs. direct or video laryngoscopy. Can you please elaborate on this? This is the proper comparator for a failed airway. 

Thanks for your comments. In our review, we found four studies that compared SGDs with direct laryngoscopy (Kurowski, Shavit, Bielski, Gawlowski) and in two the comparison was made with video laryngoscopy (Liu, Fun); however, the objective of our review was to evaluate the success rate and blind intubation time with SGDs, as reflected in Table 1. Considering the above, from these six studies in question, only the outcome information of the group that used SGDs for blind intubation purposes was extracted.

Discussion:

- Comment #1: Differences in intubation time: Please note that the differences can be due to the definitions used for “intubation time.” Were you able to extract the definition from the respective studies? There is a big difference when the intubation time is measured from positioning the ET tube through an already placed LMA compared to when the LMA has to be placed and removed afterward. This should be a part of your discussion and can explain fast intubation times (4 seconds!). Reference 40: “Times for tracheal intubation were measured from the beginning of first insertion of the laryngeal mask conduit to completion of tracheal intubation. This timing included the time required for removal of the laryngeal mask conduit, confirmation of ventilation with capnography, and use of the Macintosh laryngoscope where intubation with the LMA Fastrach ™ or LMA CTrach ™ systems had failed.”

Reference 28: “Insertion time (s) (defined as the time taken from the moment the ETT was inserted into the SGA device till an adequate EtCO2 trace was obtained).”

If this is not accounted for in your discussion, the blind intubation time should be interpreted cautiously. If there is too much heterogeneity in the definitions used, this should be stated as well.

We agree with the reviewer. We have clarified the differences in intubation time in the results and in the discussion. Regarding the intubation time, it was heterogeneous between the studies since this outcome can be evaluated in different ways, such as tube passage time or SGD placement time.

- Comment #2: “The LMA Fastrach and i-gel exhibited the highest first-attempt success rates, followed by iLTS-D2 and S.A.L.T., all surpassing an 80% success rate. These results are satisfactory and promising from a clinical perspective.” A rate of 80% is not satisfactory. Current resuscitation guidelines state that an expert needs a 95% success rate within two attempts. This should be the standard against which the SGAs are compared. Please revise your discussion accordingly. 

Thanks for the precision, we agree to eliminate that part of the discussion.

- Comment #3: “While it is crucial for an anesthesiologist to establish a definitive airway rapidly, maintaining adequate patient ventilation is equally important regardless of the attempts made.” It is not only crucial for anesthesiologists. During an elective induction, there is always the (theoretical) possibility of reversing the anesthesia. This is not possible in an emergency setting as the indication for airway management is prevention of hypoxia and death.

Thank you for your valuable comment, we agree with what you mention, and we have improved the writing of the text.

- Comment #4: There is only one newer study (Zuercher, 2022, which included patients from January 2017 to July 2019!). How has the evolution of video laryngoscopy and the use of styles or bougies affected the relevance of an intubation LMA? Is this still used in routine practice? You could look for recommendations for difficult airway carts (e.g. https://www.ncbi.nlm.nih.gov/pmc/articles/PMC6369510/)

Thank you for your valuable comment. We agree with what you mentioned and have improved the wording of the text.

Conclusion:

- Comment #1: Please shorten your conclusion. The first two sentences can be omitted.

Thanks for your comment. We have introduced the suggested changes in the manuscript. 

Reviewer #2: Comments

- Comment #1: I think this is an important topic which will be very relevant for clinical application. I commend the authors for the hard work they have put into this manuscript. I appreciate the time taken to revise this manuscript from the initial submission to this current submission as a systematic review. I think it is well written, addresses the primary and secondary outcomes

Thanks for your comment. We have introduced all the suggested changes in the manuscript and in our opinion, the quality of the manuscript has increased.

Reviewer #3: Comments

- Comment #1: The topic of using supraglottic airway devices remains relevant despite the increasing use of video laryngoscopes, which do not provide an absolute solution for all patients. For safety reasons, skills in emergency intubation with supraglottic devices should be maintained, and it is important to compare the available devices to choose the most suitable one. The presented study is interesting and well-written. However, the data is heterogeneous, including studies on patients, mannequins, and cadavers, which creates some confusion and makes a meta-analysis difficult.

We appreciate your evaluation. In this new version, we excluded one study that used cadaver. In this sense, the results were rewrites. Additionally, in the discussion section we have added like limitation the methodological heterogeneity across studies therefore a meta-analysis could not be performed.

- Comment #2: The feedback from previous reviewers has been well incorporated into the new text. However, the inclusion of a single study involving cadavers may introduce more bias than benefit to the analysis and could be excluded to limit the focus to studies involving patients and mannequins, which already exhibit some heterogeneity.

Thanks for your comment. We followed your recommendation, and we excluded the study with cadavers. In this sense, the results were rewrites.

- Comment #3: Regarding the populations performing supraglottic device intubation, it appears they were individuals already trained in using the device rather than participants in a training study. This clarification would be valuable for interpreting the results.

Thank you for your precision. We agree with what you mention, and we have clarified it in the manuscript.

- Comment #4: The distribution of devices used seems well represented across continents, with no clustering of studies on a single model in any country or continent. This could also be mentioned in the discussion to reinforce the universality of the message.

Thank you for your valuable comment. We have added this information to the discussion.

---

## [Decision Letter · Decision Letter 1]

25 Nov 2024

Supraglottic airway devices for blind endotracheal intubation: A systematic review.

PONE-D-24-29958R1

Dear Dr. Jorman H. Tejada,

We’re pleased to inform you that your manuscript has been judged scientifically suitable for publication and will be formally accepted for publication once it meets all outstanding technical requirements.

Kind regards,

Silvia Fiorelli

Academic Editor

PLOS ONE

Reviewers' comments:

Reviewer's Responses to Questions

**Comments to the Author**

1. If the authors have adequately addressed your comments raised in a previous round of review and you feel that this manuscript is now acceptable for publication, you may indicate that here to bypass the “Comments to the Author” section, enter your conflict of interest statement in the “Confidential to Editor” section, and submit your "Accept" recommendation.

Reviewer #1: All comments have been addressed

Reviewer #2: All comments have been addressed

2. Is the manuscript technically sound, and do the data support the conclusions?

Reviewer #1: Yes

Reviewer #2: Yes

3. Has the statistical analysis been performed appropriately and rigorously? 

Reviewer #1: Yes

Reviewer #2: Yes

4. Have the authors made all data underlying the findings in their manuscript fully available?

Reviewer #1: Yes

Reviewer #2: Yes

5. Is the manuscript presented in an intelligible fashion and written in standard English?

Reviewer #1: (No Response)

Reviewer #2: Yes

6. Review Comments to the Author

Reviewer #1: (No Response)

Reviewer #2: Thank you to the authors for submitting a revised version of their manuscripts and responding to all comments. I think it is an interesting topic and this manuscript adds to this topic.

7. PLOS authors have the option to publish the peer review history of their article (what does this mean?). If published, this will include your full peer review and any attached files.

Reviewer #1: **Yes: **Stephan Katzenschlager

Reviewer #2: No

---

## [Editor Report · Acceptance letter]

28 Nov 2024

PONE-D-24-29958R1 

PLOS ONE

Dear Dr. Tejada-Perdomo, 

I'm pleased to inform you that your manuscript has been deemed suitable for publication in PLOS ONE. Congratulations! Your manuscript is now being handed over to our production team.

Kind regards, 

on behalf of

Dr. Silvia Fiorelli 

Academic Editor

PLOS ONE